# Exosomes Derived from Hypoxia-Cultured Human Adipose Stem Cells Alleviate Articular Chondrocyte Inflammaging and Post-Traumatic Osteoarthritis Progression

**DOI:** 10.3390/ijms241713414

**Published:** 2023-08-29

**Authors:** Ling-Hua Chang, Shun-Cheng Wu, Chung-Hwan Chen, Jhen-Wei Chen, Wan-Chun Huang, Che-Wei Wu, Yi-Shan Lin, Yu-Ju Chen, Je-Ken Chang, Mei-Ling Ho

**Affiliations:** 1Regenerative Medicine and Cell Therapy Research Center, Kaohsiung Medical University, Kaohsiung 807, Taiwan; linghua_chang@yahoo.com.tw (L.-H.C.); shunchengwu@hotmail.com (S.-C.W.); hwan@cc.kmu.edu.tw (C.-H.C.); h90085663@gmail.com (J.-W.C.); wanchun1010@gmail.com (W.-C.H.); tkdiven@gmail.com (C.-W.W.); 327lin@gmail.com (Y.-S.L.); yujutw1231@gmail.com (Y.-J.C.); jkchang@cc.kmu.edu.tw (J.-K.C.); 2Orthopaedic Research Center, Kaohsiung Medical University, Kaohsiung 807, Taiwan; 3Department of Nursing, Asia University, Taichung 41354, Taiwan; 4Department of Orthopaedics, College of Medicine, Kaohsiung Medical University, Kaohsiung 807, Taiwan; 5Department of Orthopaedics, Kaohsiung Municipal Ta-Tung Hospital, Kaohsiung Medical University, Kaohsiung 807, Taiwan; 6Graduate Institute of Medicine, College of Medicine, Kaohsiung Medical University, Kaohsiung 807, Taiwan; 7Department of Physiology, College of Medicine, Kaohsiung Medical University, No. 100, Shih-Chuan 1st Road, Kaohsiung 807, Taiwan; 8Department of Medical Research, Kaohsiung Medical University Hospital, Kaohsiung 807, Taiwan; 9Department of Marine Biotechnology and Resources, National Sun Yat-sen University, Kaohsiung 807, Taiwan; 10College of Professional Studies, National Pingtung University of Science and Technology, Pingtung 908, Taiwan

**Keywords:** exosomes, micro ribonucleic acid (miRNA), articular chondrocytes, hypoxia, inflammaging, SASP (senescence-associated secretory phenotype), osteoarthritis (OA)

## Abstract

Osteoarthritis (OA) is the most common age-related degenerative joint disease. Inflammaging, linking inflammation and aging, is found in senescent cells with the secretions of matrix-degrading proteins and proinflammatory cytokines. The senescence-associated secretory phenotype (SASP) plays a very important role in OA progression. However, there remains no effective way to suppress OA progression, especially by suppressing inflammaging and/or the chondrocyte SASP. Recent studies have shown that exosomes derived from hypoxia-cultured BMSCs can regenerate cartilage in OA animal models. Some reports have further indicated that exosomes secreted from MSCs contribute to the efficacy of MSC therapy in OA. However, whether hypoxia-cultured ADSC-secreted exosomes (hypoxia-ADSC-Exos) can alleviate the chondrocyte SASP or OA progression remains unclear. Accordingly, we hypothesized that hypoxia-ADSC-Exos have a beneficial effect on the normal functions of human articular chondrocytes (HACs), can attenuate the SASP of OA-like HACs in vitro, and further suppress OA progression in rats. Hypoxia-ADSC-Exos were derived from ADSCs cultured in 1% O_2_ and 10% de-Exo-FBS for 48 h. The molecular and cell biological effects of hypoxia-ADSC-Exos were tested on IL1-β-induced HACs as OA-like HACs in vitro, and the efficacy of OA treatment was tested in ACLT-induced OA rats. The results showed that hypoxia-ADSC-Exos had the best effect on GAG formation in normal HACs rather than those cultured in normoxia or hypoxia plus 2% de-Exo-FBS. We further found that hypoxia-ADSC-Exos alleviated the harmful effect in OA-like HACs by decreasing markers of normal cartilage (GAG and type II collagen) and increasing markers of fibrous or degenerative cartilage (type I or X collagen), matrix degradation enzymes (MMP13 and ADAMT5), and inflammatory cytokines (TNFα and IL-6). More importantly, intra-articular treatment with hypoxia-ADSC-Exos suppressed OA progression, as evidenced by the weight-bearing function test and cartilage GAG quantification in ACLT rats. Moreover, through NGS and bioinformatic analysis, seven potential miRNAs were found in hypoxia-ADSC-Exos, which may contribute to regulating cellular oxidative stress and attenuating cell senescence. In summary, we demonstrated that hypoxia-ADSC-Exos, carrying potent miRNAs, not only improve normal HAC function but also alleviate HAC inflammaging and OA progression. The results suggest that hypoxia-ADSC-Exo treatment may offer another strategy for future OA therapy.

## 1. Introduction

Osteoarthritis (OA) is the most common age-related degenerative joint disease with chronic inflammation [1]. The progressive destruction of articular cartilage is one of the hallmarks of OA, with loss of proteoglycan and collagen type II (two of the most prevalent components of normal hyaline cartilage), subchondral bone sclerosis, osteophyte formation, and synovial hyperplasia [2,3,4,5]. There remains a clinically unmet need to prevent, stop, or even restrain the progression of OA. Promising novel pharmacological or cell-related treatments are thus urgently important.

Inflammaging, chronic, sterile, and low-grade inflammation during aging, has been reported to contribute to the pathogenesis of age-related diseases [6]. Moreover, senescent cells secrete matrix-degrading proteins and proinflammatory cytokines, called the “senescence-associated secretory phenotype (SASP)”. The SASP is associated with the production of proinflammatory cytokines (IL-6, IL-1α/β, etc.), chemokines (MCP-1, IL-8, etc.), growth factors (TGF-β, VEGF, etc.), and proteases (MMP-3, MMP13, etc.) [7]. Although OA is a degenerative arthritis, low-grade inflammation appears during the OA process. The levels of several inflammatory mediators, such as IL6, IL1-β, and MCP1, were also observed to be higher in OA sera than in healthy sera [8]. Accordingly, chondrocyte changes during OA progression are highly linked with inflammaging. Among the inflammatory mediators, IL-1β was found to be higher in the extracellular matrix of OA chondrocytes, especially surrounding the superficial zone [9]. In addition, IL1-β was also demonstrated to drive chondrocytes toward a senescent phenotype and induce the secretion of SASP factors [10]. Therefore, IL1-β-induced senescence in chondrocytes has been used as an in vitro OA-like cell model [11].

Mesenchymal stem cells (MSCs), especially bone marrow MSCs (BMSCs) and adipose-derived MSCs (ADSCs), have been intensively investigated for cell-based treatment of cartilage lesions and OA in animal [12,13,14] and human studies [15,16,17]. Using ADSCs as the MSC source may be beneficial because a large number of ADSCs can be obtained from a relatively small amount of fat tissue [18,19], and the proliferation potential of ADSCs does not appear to decline with donor age [20]. On the other hand, much evidence has shown that the efficacy of MSC therapy in OA does not result from the chondrogenesis of MSCs but relies on the paracrine effect of MSCs [21]. The paracrine effect on tissue regeneration, immune regulation, and anti-inflammation may be from the secretome of MSCs, including growth factors, cytokines, and extracellular vesicles (EVs) [22,23,24].

Exosomes are EVs with a size of approximately 40 to 160 nm that contain constituents from a cell, including DNA, RNA, lipids, metabolites, and cytosolic and cell-surface proteins [25]. The function of exosomes is to mediate cell-to-cell communication by carrying various biologically active molecules [26]. Studies have indicated that exosomal miRNAs play fundamental roles in cell-to-cell communication in both healthy and pathological conditions [27,28]. More importantly, recent studies indicated that MSC-derived exosomes could promote cartilage regeneration in OA animal studies [29,30]. However, it has not been well investigated whether ADSC-derived exosomes have an anti-inflammaging effect on articular chondrocytes.

Oxygen is an important component of the cellular microenvironment, serving as both a metabolic substrate and a signaling molecule. Usually, in vitro cell/tissue cultures are maintained at an atmospheric oxygen level of approximately 20% (160 mmHg); however, normal tissues exist in the body under physiological hypoxic conditions with approximately 2–9% O_2_ (14–65 mmHg), which varies by location. Accordingly, MSCs exist in a physiological hypoxic environment in the body [31,32]. In addition, articular cartilage is avascular and receives nutrients and oxygen, mainly from the surrounding synovial fluid. The living environment of chondrocytes is approximately 6% O_2_ (45 mm Hg) in the superficial zone and less than 1% O_2_ (7.6 mm Hg) in the deep zone [33]. Previous studies found that hypoxia pretreatment of human ADSCs led to better potential for chondrogenic differentiation [34,35]. Recent studies have shown that exosomes derived from hypoxia-cultured BMSCs can regenerate cartilage in OA animal models [36]. However, whether hypoxia-cultured ADSC-secreted exosomes (hypoxia-ADSC-Exos) can suppress SASP and thus alleviate OA progression remains unclear. Accordingly, we hypothesize that hypoxia-ADSC-Exos have a beneficial effect on the normal functions of HACs, attenuate the SASP of OA-like HACs in vitro, and further suppress OA progression in rats. Another important issue is to search for effective molecular mechanisms of ADSC-Exos on articular chondrocytes. The miRNA profiles between hypoxia-ADSC-Exos and normoxia-ADSC-Exos will also be investigated.

## 2. Results

### 2.1. Optimal Culture Conditions for ADSC-Exos to Promote Human Articular Chondrocyte Functions

To explore the effect of different induced ADSC-Exos (2N, 2H, 10N, 10H) on cultured HACs, we first assessed the characteristics of ADSC-Exos by TEM, Western blotting, and NTA. Second, we analyzed the effect of four different induced ADSC-Exos on the biological activity of HACs by testing the uptake of exosomes into chondrocytes and the subsequent HAC function. The results show that ADSC-Exos were successfully isolated from the CM of ADSCs. ADSC-Exos displayed a rotundity shape by TEM (Figure 1A). Western blot analysis of ADSC-Exo lysates showed positive expression of the exosome surface markers CD9, CD63, and CD81; cytosolic/lipid-bound protein markers ALIX and TSG101; and one negative marker, α-tubulin (Figure 1B). The NTA showed that the average vesicle diameter was approximately 130 nm, and the cell sizes showed no significant difference among the four groups (Figure 1C). Moreover, the exosome concentration results showed that ADSCs in hypoxic and rich nutrition environments could secrete more abundant exosome particles than the other groups (Figure 1D). To examine the cell biological effects of ADSC-Exos on HACs, the uptake of exosomes by HACs was tested. The chondrocytes were prelabeled with CellTracker (green) and incubated with CM-Dil-labeled exosomes (red) for 12 days. Exosomes were observed in the cytoplasm to confirm uptake by HACs, and the results only showed 10H-Exos uptake (Figure 1E). To determine the optimal functional effects of four differentially induced ADSC-Exos (2N, 2H, 10N, 10H) on normal human articular chondrocytes (HACs), we compared HAC function by using Alcian blue staining and the DMMB assay to evaluate GAG synthesis. We also used HACs cultured in a culture medium containing 0.1% insulin, transferrin, and selenium (ITS) and 10% FBS as a positive control and HACs cultured in a basal medium without ITS and FBS as a negative control. The results showed that HACs treated with ADSC-Exos (10H) exhibited the bluest color by Alcian blue staining (Figure 2A), and the quantification results were similar in the DMMB assay. Four condition-induced ADSC-Exos groups had significantly better effects than the basal control group, but only the 10H group had significantly better effects than the positive control group (Figure 2B). These results demonstrated that four differently induced ADSC-Exos could improve the ability of chondrocytes to deposit GAG in vitro, and hypoxia with 10% de-Exos-FBS (10H) had the optimal effect. The dose of 10H-ADSC-Exos (10^7^~10^9^ particles/mL) did not alter the cell viability of HACs (Figure 2C); however, 10H-ADSC-Exos (10^7^~10^9^ particles/mL) dose-dependently increased chondrocyte proteoglycan deposition (Figure 2D). Therefore, subsequent experiments used 10H exosomes (1 × 10^9^ particles/mL) to treat OA-like chondrocytes and an OA rat model.

### 2.2. Hypoxia-Induced ADSC-Exos Enhanced Cartilaginous Matrix Synthesis and Inhibited Fibrous/Degenerated Cartilage in IL-1β-Induced OA-like Inflammaging Articular Chondrocytes

IL-1β drives chondrocytes toward a senescent phenotype, and it is the most widely used cytokine to induce an OA-like phenotype in chondrocytes in vitro [11]. Accordingly, exposure to IL-1β stimulates chondrocytes to secrete SASPs, such as IL-6, IL-8, MMP-13, and the resulting degradative cascade. Therefore, we added different doses of IL-1β (2, 5, 10 ng/mL) for 2 days to stimulate HACs. The results showed dose-dependent increases for the mRNA expression of COX-2, IL6, and MMP13 and dose-dependent inhibition for the mRNA expression of aggrecan (Appendix A). Next, we chose the highest dosage of IL-1β (10 ng/mL) treatment for 12 days to verify chondrocyte senescence by detecting senescence-associated β-galactosidase (SA-β-gal), which is the most commonly used senescence marker [37]. The results showed that IL1-β treatment resulted in more SA-β-gal-positive cells than the control (Appendix A), suggesting that IL-1β caused chondrocyte inflammaging. We used this OA-like cell model to evaluate the efficacy of hypoxia-ADSC-Exos (10H) in the treatment of OA to assess normal functional genes in chondrocytes. The results showed that hypoxia-ADSC-Exo treatment increased the mRNA expression of aggrecan, col2a1, and PTHrP in IL-1β-treated inflammaging chondrocytes, especially aggrecan (>3-fold changes) (Figure 3A). Similar findings were also demonstrated in the protein level of type II collagen and GAG content detected by ELISA, Alcian blue staining, and DMMB assay. Hypoxia-ADSC-Exos dose-dependently increased normal functional matrix (hyaline cartilage) synthesis in OA-like inflammaging chondrocytes (Figure 3B–D). In contrast to hyaline cartilage, if the articular cartilage surface is impaired, type I collagen will be formed in early osteoarthritis [38]. Hypoxia-ADSC-Exos significantly suppressed the mRNA expression and protein level of type I collagen in IL-1β-treated chondrocytes (Figure 4A,C). Type X collagen, a degeneration and hypertrophy marker, showed similar results in that hypoxia-ADSC-Exos significantly inhibited its mRNA expression and protein levels in a dose-dependent manner (Figure 4B,D). In summary, these findings indicate that hypoxia-ADSC-Exos show therapeutic efficacy in inflammaging articular chondrocytes.

### 2.3. Hypoxia-Induced ADSC-Exos Suppress Inflammatory Cytokines and Degradation Enzymes in IL-1β Induced OA-like Inflammaging Articular Chondrocytes

Since inflammation causes cartilage degeneration, we explored the effect of hypoxia-ADSC-Exos on SASP secretion in OA-like inflammaging chondrocytes. Then, we assessed the mRNA expression of the inflammatory-related genes CEBPβ, COX-2, IL-6, and TNF-α and catabolic genes MMP-13 and ADAMT5. The results showed that IL-β upregulated these genes, but hypoxia-ADSC-Exo treatment significantly suppressed them (Figure 5A,B). We also collected the conditioned medium and verified the protein levels of IL-6, TNF-α, and MMP13 via ELISA (Figure 5C–E). The protein quantification results showed that IL1-b dramatically upregulated IL-6 secretion (>400 pg/mL) after 2 days of treatment, but hypoxia-ADSC-Exos significantly reduced IL-6 secretion after 4 days of treatment (Figure 5C). In addition, ELISA results indicated that hypoxia-ADSC-Exos had more efficient effects on suppressing TNF-α, and the protein concentration of TNF-α was even less than that in the control groups (Figure 5D). MMP13 ELISA showed similar results to IL-6 (Figure 5E). These findings indicated that hypoxia-ADSC-Exos have anti-inflammatory and antidegradation effects on OA-like articular chondrocytes.

### 2.4. Hypoxia-Induced ADSC-Exos and ADSC Treatment Alleviate ACLT-OA Progression in Rats

To compare the chondroprotective effect of hypoxia-ADSC-Exos and ADSCs in vivo, a rat ACLT-OA model was used. The study design is shown in Figure 6A; after 2 weeks of surgery, 40 µL of PBS (sham Ctrl and OA group), 40 µL of ADSCs 1 × 10^6^ cells in PBS (OA+ADSCs group), and 40 µL of hypoxia-ADSC-Exo 7.5 × 10^9^ particles in PBS (OA+ hypoxia-ADSC-Exo group) were IA injected into the right knee of rats. The OA+ADSC group received only one injection, but the other groups were injected with PBS or hypoxia-ADSC-Exos weekly (a total of seven administrations). In addition, a weight-bearing test was performed before and after OA surgery and combined with hypoxia-ADSC-Exo or ADSC treatment to evaluate the exercise function of the knee. Three weeks after surgery, the sham Ctrl group was restored to nonsurgery conditions; however, the OA group could not recover the bearing weight function and was significantly decreased compared to the sham Ctrl group. On the other hand, administration of hypoxia-ADSC-Exos and ADSCs gradually restored knee-bearing function, especially at 9 weeks after surgery, which was not significantly different from the sham Ctrl group. A comparison of the OA+Exo group and the OA+ADSC group with the OA group showed a significantly increased percentage of weight distribution between weeks 4 and 9 after ACLT surgery. Comparing the OA+Exo and OA+ADSC groups showed no significant difference, but the OA+Exo group had a slightly better weight distribution than the OA+ADSC group (Figure 6B). Therefore, these results demonstrate that administration of hypoxia-ADSC-Exos and ADSCs improved the knee function of ACLT-OA in weight-bearing tests and may be beneficial for pain relief. The articular cartilage surface of the tibia presented an intact macroscopic appearance with no sign of lesions in the sham ctrl group, but fibrillation and erosion occurred in the tibial plateau of the ACLT-OA group. The OA+Exo group showed less erosion and fibrotic cartilage than the ACLT-OA group, and the OA+ADSC group showed a thicker cartilage surface (Figure 6C). In cartilage sections that were stained with safranin O-fast green to evaluate proteoglycan synthesis and quantified by red area density, a significant loss of proteoglycan was observed in the ACLT-OA group compared with the sham Ctrl group. The OA+Exo group and the OA+ADSC group significantly increased GAG deposition compared with the ACLT-OA group (Figure 6D). ADAMTS5 is a member of the ADAMTS protein family and functions as an aggrecanase to cleave aggrecan. IHC-stained ADAMT5 represents an opposite pattern to GAG staining (red). The ADAMT5 increase in the cartilage of the ACLT-OA group is equal to the green color of safranin O-fast green stain, which causes GAG loss. The OA+Exo group and the OA+ADSC group had lower ADAMT5 levels than the ACLT-OA group (Figure 6G). IHC staining of type II collagen indicated a surface loss in the ACLT-OA group, and the sham Ctrl, OA+Exo, and OA+ADSC groups showed no type II collagen loss (Figure 6E). Interestingly, we found no cells or nuclei stained on the surface of thick cartilage in the OA+ADSC group. Therefore, we next evaluated the level of MMP13 in cartilage, a key enzyme in the cleavage of type II collagen, which plays a pivotal role in the breakdown of cartilage in osteoarthritic joints. MMP13 is primarily expressed in the cartilage surface, which is equal to the loss of type II collagen in the ACLT-OA group, and IL-1β has similar results. The OA+Exo group showed moderately reduced MMP13 and IL1-β levels, but the OA+ADSC group showed no obvious suppression (Figure 6F,H). However, IL1-β and CD68 macrophage markers showed very mild elevation and no significant difference in the synovium among the four groups. Taken together, hypoxia-ADSC-Exos and ADSCs have a chondroprotective effect that suppresses cartilage erosion caused by ADAMT5, MMP13, and IL1β and reverses proteoglycan and type II collagen in OA cartilage.

### 2.5. Next-Generation Sequencing (NGS) and Bioinformatic Analysis of miRNA Profiles in Hypoxia-Induced ADSC-Exos

To identify the hypoxia-related exosomal miRNAs expressed in ADSCs, we used four culture conditions (2H, 2N, 10H, 10N) to induce ADSC-Exos and small RNA sequencing analysis to identify the exosomal miRNAs. There was differential expression of 136, 58, 35, and 24 miRNAs under the 2H, 2 N, 10H, and 10N conditions, respectively, by screened NGS data (*p* < 0.05). The intersection numbers of different exosomal miRNAs were presented in the four groups (Figure 7A). Heatmap analysis showed the highly expressed exosomal miRNAs ranked at 10H, and the miRNA expression profile decreased from the highest level to ~1000 RPM. Then, we compared the exosomal miRNA expression between 10N, 2H, and 2N with 10H; the highly expressed miRNA was represented in red, and conversely, the lowest was represented in blue (Figure 7B). We found several of the highest exosomal miRNAs expressed in 10H and 2H, but they also simultaneously expressed some of the lowest exosomal miRNAs. Therefore, we selected seven hypoxia-related miRNAs, including miR-381-3p, miR-122-5p, miR-143-3p, miR-206, miR-30a-5p, miR-22-3p, and miR-151a-5p, to predict the miRNA-target gene network. A total of 3358 possible target genes were predicted by using TargetScan v7.2, miRanda, and miRDB to perform Gene Ontology (GO) and Kyoto Encyclopedia of Genes and Genomes (KEGG) enrichment analyses. GO functions were conducted to analyze the molecular function, cellular component, and biological process of these target genes. The GO functions are shown in Figure 7C and include cell growth, response to oxidative stress and oxygen levels, cell-substrate adhesion, and chondrocyte development. The KEGG pathway enrichment analysis was used to investigate the pathways of these target genes, and the highest correlated pathways included the MAPK, Rap1, cGMP-PKG, Ras, PI3K-Akt, cellular senescence, and immune-related signaling pathways (Figure 7D). These miRNAs and their target genes are potentially related to OA treatment and require further study. These findings proposed an underlying mechanism of hypoxia-ADSC-Exos in OA cartilage protection through miRNA to inhibit SASP secretion.

## 3. Discussion

MSC-based OA treatment has become a new approach. Recently, MSC-derived exosomes have emerged as a prominent candidate. One of the most significant challenges is to obtain better quality exosomes for optimal efficacy in OA treatment. In this study, we demonstrated that exosomes derived from hypoxia (1% O_2_) and 10% de-Exo-FBS medium cultured ADSCs showed the best effect on HAC function compared to those cultured in normoxia or hypoxia plus 2% de-Exo-FBS. This result indicated that the culture conditions of ADSCs mimicking their physiological niche could release better quality ADSC-Exos. More importantly, we found that hypoxia-ADSC-Exos significantly suppressed the SASP, including degradation enzymes, inflammatory cytokines, and markers of degenerative cartilage, in IL-1β-induced OA-like HACs. This result indicated that hypoxia-ADSC-Exos alleviated the inflammaging of senescent articular chondrocytes. Furthermore, in this study, we first demonstrated that intra-articular treatment with hypoxia-ADSC-Exos alleviated OA progression in ACLT-OA rats, as evidenced by improved knee weight-bearing function, articular cartilage integrity, and normal matrix levels (collagen type II and GAG), as well as decreased matrix degradative enzymes (MMP13 and ADAMPs) and the inflammatory mediator IL-1β. The findings from this study demonstrated that ADSCs cultured in 1% O_2_ hypoxic conditions induced more effective exosomes that suppressed the inflammaging of senescent articular chondrocytes and thus alleviated OA progression in vivo.

MSCs are known to support differentiated cells in a tissue when needed. The microenvironment provides crucial niche factors for both cell types. One of the important niche factors is oxygen tension, which is much lower than that in the atmosphere. The physiological tissue oxygen tension, approximately 1–9%, is rather hypoxic compared to the 20% oxygen tension found in the atmosphere. Even though those MSCs are located close to vascular niches, their oxygen tension ranges from 1–8% [39]. Therefore, it is proposed that MSCs require a hypoxic environment to maintain an undifferentiated status and a relative advantage. On the other hand, hypoxia-pretreated human MSCs showed a better effect on osteogenic or chondrogenic differentiation [34,35]. This result indicated that a hypoxic microenvironment is beneficial for MSCs by preventing destruction, maintaining stemness, and maintaining multipotent differentiation potential [40]. Furthermore, it was also reported that stem cell-secreted bioactive molecules, trophic factors, or immunomodulatory factors may be transported by exosomes to help neighboring cells overcome stress or obtain beneficial properties [41]. Another possible effect of hypoxia-MSCs may be to release a beneficial secretome for the surrounding differentiated cells to maintain normal function or even rescue senescence. In this study, we demonstrated that hypoxia-ADSC-derived exosomes had a better effect on maintaining the normal function of differentiated articular chondrocytes. More importantly, the hypoxia-ADSC-derived exosomes further significantly alleviated inflammaging in IL1β-induced senescent HACs by reducing their SASP.

Several clinical trials using intra-articular (IA) injections of MSCs to treat OA have achieved certain therapeutic effects; however, some studies have shown unsatisfactory results [42]. Usually, MSC therapy uses only one IA injection. A previous study to trace IA-injected GFP-MSCs showed that only 1.6% of MSCs remained in the knee joint of OA mice [43]. Therefore, the efficacy of MSC therapy on OA may rely on MSC-secreted growth factors, cytokines, or exosomes to initiate the repair process but may not rely on MSC differentiation. Accordingly, in this study, we treated hypoxia-ADSC-exosomes with weekly IA injections seven times, mimicking the continuous secretion from ADSCs in joints, and compared the efficacy with the one-time IA injection of ADSCs simultaneously. The results revealed that both treatments of hypoxia-ADSC-exosomes and ADSCs have similar effects on the weight-bearing function, preserving the cartilage matrix of GAG and collagen type II, as well as decreasing matrix degradative enzymes (ADAMT5 for aggrecan and MMP13 for collagen type II) and the inflammatory mediator IL1β. The weight-bearing function recovered from ACLT surgery was slightly faster in the hypoxia-ADSC-exosome group than in the ADSC group (Figure 6B). The appearance of the cartilage surface was smoother in the hypoxia-ADSC-exosome group (Figure 6C). In cartilage, histological sections showed more superficial area lost lacunae and cells in the ADSC group (Figure 6E–H). The findings from the OA animal study indicated that the efficacy of hypoxia-ADSC exosomes in improving OA symptoms may be similar to that of ADSC treatment, but the mechanism may not be the same. It is possible that the hypoxia-ADSC-exosomes carry more effective factors to preserve cartilage chondrocytes and matrix by suppressing SASPs.

Some limitations remained in the in vitro and in vivo OA models in this study. In vitro cell biological OA models only mimic chondrocytes responding to IL-1β; however, during OA progression, chondrocytes undergo terminal differentiation, receiving multiple signal changes from growth factors and inflammatory cytokines. Nevertheless, the IL-1β induction model was able to express SASPs and was further suppressed by hypoxia-ADSC-Exos. It is suggested to be used to test the effect of hypoxia-ADSC-Exos on OA-like HACs in vitro prior to subsequent in vivo study. For the ACLT-OA animal model, post-traumatic osteoarthritis (PT-OA) cannot represent spontaneous aging caused by OA. However, PT-OA is also very significant because it more frequently occurs in young people to accelerate cartilage degeneration. In this study, we demonstrated that the OA change in PT-OA cartilage is alleviated by hypoxia-ADSC-Exo treatment. The efficacy of the other causes of OA also needs to be further tested before clinical study. Additionally, the ACLT-OA rat experiment was stopped at 9 weeks after ACLT surgery, and the cartilage damage as well as synovium inflammation were still mild as in early-stage OA. More effects of hypoxia-ADSC-Exo treatment on cartilage and synovium in moderate or severe OA need to be examined in future animal studies.

Exosomal miRNAs have been indicated to be very important for transporting intercellular communication signals [27,28]. MSC-exosomes have been reported to improve cartilage regeneration in OA animals [29,30,44]. Indeed, several specific miRNAs were found to be differentially expressed in normal and OA cartilage; however, whether they contribute to OA progression has not been investigated [45]. A study indicated that hADSC-Exos are rich in several miRNAs (i.e., miR-199a, miR125b, miR-221, and miR92a) that may be related to inflammation in OA [30]. How those exosomal miRNAs contribute to regenerating articular chondrocytes remains unclear. In this study, we compared the exosomal miRNA profiles of ADSC-Exos between normoxia and hypoxia and found that seven miRNAs, miR-381-3p, miR-122-5p, miR-143-3p, miR-206, miR-30a-5p, miR-22-3p, and miR-151a-5p, were more abundant in hypoxia-ADSC-Exos than in normoxia-ADSC-Exos. Further, GO and KEGG analysis showed that these target genes may be involved in cell growth, response to oxidative stress, cell-substrate adhesion, chondrocyte development, cellular senescence and immune signaling pathways.

Among these seven hypoxia-related miRNAs, a previous study has demonstrated that miR-30a protects aged cardiomyocytes from injury [46]. The miR-206 was indicated to regulate several significant biological functions and play a key modulator role in skeletal muscle development and disease, such as promoting extracellular matrix accumulation by targeting DNMT3A [47]. In addition, miR-22-3p was reported to target oxidative stress and mitochondrial dysfunction pathways to induce a protective role in acute kidney injury [48]. The miRNA profiles in ADSC-exosomes and the difference between hypoxia-ADSC-Exos and normoxia-ADSC-Exos in this study, as well as those from previous studies, support very important information for further investigation of the molecular mechanism to clarify how hypoxia-ADSC-Exos alleviate the inflammaging of HACs and OA progression.

Looking back, traditional treatments for OA include the management of pain and inflammation by anti-inflammatory drugs or, in several cases, even corticosteroid injections. At the end-stage of OA, joints need replacing. In current OA treatments, despite the administration of NSAIDs, viscosupplementation, a treatment involving hyaluronan intra-articular injections, has also been used, but it is not effective for those with damaged articular cartilage [49]. For mild to moderate OA, several treatment strategies of regenerative medicine are explored, such as using platelet-enriched plasma (PRP), cell-based therapy or acellular MSC exosomes. In the present study, we demonstrated that using hypoxia-ADSC-Exos can alleviate the degeneration of articular chondrocytes and eventually OA progression, which may be potentially a future treatment method for OA.

In summary, we demonstrated that hypoxia-ADSC-Exos not only improve normal HAC function but also alleviate HAC inflammaging and OA progression. The seven potential miRNAs carried by hypoxia-ADSC-Exos may contribute to executing the function. Hypoxia-ADSC-Exo treatment may offer another strategy for future OA therapy.

## 4. Materials and Methods

### 4.1. Isolation and Culture of Human Adipose-Derived Stem Cells (hADSCs)

For the isolation of ADSCs from humans, hADSCs were isolated from subcutaneous adipose tissue as previously described [50]. Human adipose tissue was obtained from patients with approval from the ethical committee at the Kaohsiung Medical University Hospital (IRB number: KMUH-IRB-E(II)-20150193). After obtaining informed consent, subcutaneous adipose tissue from the gluteal area was taken from patients during plastic surgery. After 3 g of subcutaneous adipose tissue was isolated from humans, it was minced with scissors. The minced adipose tissues were digested with 1 mg/mL of type IA collagenase (125 units/mg) at 37 °C under 5% CO_2_ for 24 h. The digested tissue was centrifuged at 1000 rpm for 5 min, and the pellet was washed twice with PBS. The K-NAC medium is composed of Keratinocyte-SFM Basal Medium (Gibco-BRL, Rockville, MD, USA) supplemented with 25 mg bovine pituitary extract (BPE) (Gibco-BRL, Rockville, MD, USA), 2.5 µg human recombinant epidermal growth factor (rEGF) (Gibco-BRL, Rockville, MD, USA), 2 mM N-acetyl-L-cysteine, 0.2 mM L-ascorbic acid 2-phosphate sesquimagnesium salt, and 5% fetal bovine serum (FBS) [51]. The pellet was then resuspended in K-NAC medium and plated in a 100 mm culture dish. The K-NAC medium used in this study helped isolate and expand hADSCs, according to previous reports. In this study, we used hADSCs at passages 6–8 (P6–P8).

### 4.2. Induction and Isolation of hADSC-Derived Exosomes (ADSC-Exos)

To investigate the oxygen and serum concentration effects on the components and biological functions of ADSCs-Exo, four culture conditions were used to induce ADSCs-Exo, namely: (1) 10H: ADSC-Exos were induced under 20% O_2_ (normoxia) + 10% de-Exo-FBS (Thermo Fisher Gibco™, Waltham, MA, USA); (2) 10N: ADSC-Exos were induced under 1% O_2_ (hypoxia) + 10% de-Exo-FBS; (3) 2N: ADSC-Exos were induced under 20% O_2_ (normoxia) + 2% de-Exo-FBS; and (4) 2H: ADSC-Exos were induced under 1% O_2_ (hypoxia) + 2% de-Exo-FBS. For induction of ADSC-Exos, hADSCs were suspended in DMEM containing 1% penicillin/streptomycin and de-Exo-FBS (2% or 10%) and then plated in 100 mm culture dishes at a cell density of 1.415 × 10^4^/cm^2^ for 48 h. For hypoxic cell culture, ADSC-Exo was induced by placing hADSCs into a hypoxia incubator chamber (Catalog #27310, STEMCELL Technologies, Inc., Cambridge, UK) that was filled with a mixed gas composed of 1% O_2_, 5% CO_2_, and 94% N_2_ and then transferred into a CO_2_ incubator at 37 °C for 48 h. After 48 h, the conditioned medium (CM) in which the ADSC-Exos had been secreted was collected. ADSC-Exos were isolated from the CM by ultracentrifugation according to the protocol previously described [52]. Briefly, the CM was centrifuged at 20,000× *g* for 10 min at 4 °C to remove large cell fragments. The supernatant was then centrifuged at 50,000× *g* for 1 h to remove residual organelles and nuclei. Subsequently, it was centrifuged at 180,000× *g* for 1.5 h at 4 °C to pellet the ADSC-Exos. Thereafter, the supernatant was discarded, and the ADSC-Exos at the bottom of the centrifuge tube were resuspended in phosphate-buffered saline (PBS) to eliminate the contaminating proteins on the ADSC-Exos pellets and centrifuged at 180,000× *g* for 1.5 h at 4 °C to acquire ADSC-Exos pellets [38]. The ADSC-Exos were resuspended in 1 mL of sterilized PBS and stored at 4 °C until characterization or treatment [36]. The characterization of the ADSC-Exos involved analyses of their size, number, morphology, and protein composition [52]. The isolated ADSC-Exos stored at 4 °C were used within 2 weeks.

### 4.3. Osteoarthritis (OA)-like Chondrocyte Model and ADSC-Exos Treatment

The HACs were purchased from Clonetics^TM^ normal human chondrocytes-Knee (NHAC-kn) (Lonza, Walkersville, MD, USA). The HACs were cultured and expanded in Dulbecco’s modified Eagle’s medium (DMEM) containing 100 mg/mL ascorbic acid, 1% nonessential amino acids, 1% penicillin/streptomycin, 0.1% insulin, transferrin, and selenium (ITS) (Corning^®^ MA, New York, MA, USA), and 10% fetal bovine serum. The chondrocytes used for the OA-like cell model were in passages 7–9 (P7–P9). For induction of the OA-like cell model, chondrocytes were treated with recombinant human interleukin-1 beta (IL-1β) (Cat. no.: Z02922-10, GenScript, Piscataway, NJ, USA) at concentrations of 1 to 10 ng/mL in the culture medium.

For ADSC-Exo treatment, the ADSC-Exo solution was diluted with basal medium containing DMEM with 1% penicillin/streptomycin immediately before the start of treatment. The ADSC-Exos concentrations used in the experiments were between 10^7^ and 10^9^ particles/mL. The HACs were treated with ADSC-Exos (10^7^ to 10^9^ particles/mL) for 12 days. Three groups were examined in the present study, namely, the Ctrl group, in which the HACs were cultured in the basal medium without ADSC-Exos treatment; the IL-1β group, in which the HACs were cultured in the basal medium and treated with IL-1β at concentrations of 10 ng/mL; and the IL-1β+Exo group in which the HACs were cultured in the basal medium and treated with IL-1β and ADSC-Exos at concentrations of 10^7^ to 10^9^ particles/mL. The medium was changed every 2 days. At specific time points, the HACs were collected for further experimental analysis.

### 4.4. Nanoparticle Tracking Analysis of ADSC-Exo Size and Number

Nanoparticle tracking analysis (NTA) was performed on a ZetaView ^®^ BASIC NTA—Nanoparticle Tracking Video Microscope PMX120 (Particle Metrix GmbH, GER) equipped with a 532 nm green laser for determining the ADSC-Exos size and number. One microliter of isolated ADSC-Exos samples was diluted to 1000 µL with freshly filtered PBS (0.22 µm filter) and loaded into the detection chamber by syringe. The camera was manually set and kept the same for all samples with a slider shutter of 650 and slider gain of 50; 30 s videos were recorded, and the number of captures was 5. The detection threshold was set to 6, and blur and max jump distance were automatically set. The temperature was maintained at 25 °C. The data were processed by ZetaView software (version 8.05.14 SP7; Particle Metrix GmbH, GER).

### 4.5. Transmission Electron Microscopy (TEM)

To investigate the morphology of ADSC-Exos, transmission electron microscopy (TEM) was used. Briefly, the ADSC-Exos were dropped on a copper grid and dried in air. Then, the sample was fixed with 3% glutaraldehyde for 2 h and washed with PBS, followed by negative staining with 2% uranyl acetate for 30 s. The ADSC-Exos sample was observed using TEM (Hitachi^®^, Tokyo, Japan), and the images were captured using a digital camera (Olympus^®^, Tokyo, Japan).

### 4.6. Labeling of ADSC-Exos and Uptake by Chondrocytes

For labeling of ADSC-Exos, ADSCs were labeled with CM-DiI (V22888, Invitrogen, Carlsbad, CA, USA), and then ADSC-Exo release was induced. After 48 h of induction, the conditioned medium derived from ADSCs was collected, and CM-Dil-labeled ADSC-Exos were isolated using the protocol described in an earlier subsection through ultracentrifugation. For the uptake of ADSC-Exos by chondrocytes, the chondrocytes were seeded onto six-well plates. After 24 h, the chondrocytes were cultured for 5 days with a culture medium containing 10^9^ particles/mL CM-DiI-labeled ADSC-Exos. The medium was changed every 2 days. At specific time points, the cells were fixed with 4% paraformaldehyde in PBS for 15 min. The chondrocytes were stained with CellTracker™ Green CMFDA Dye (5-chloromethylfluorescein diacetate) (C7025, Invitrogen) for cell tracing, and their cell nuclei were stained with 4′,6-diamidino-2-phenylindole (DAPI). The uptake of the CM-DiI-labeled ADSC-Exos by the chondrocytes was visualized through confocal laser scanning microscopy (Zeiss, Weimar, GER). Images were captured using a camera.

### 4.7. Quantitative Real-Time Polymerase Chain Reaction (qRT-PCR)

Total RNA from chondrocytes was isolated using the TOOLSmart RNA Extractor (TOOLS, Taiwan). First-strand complementary DNA (cDNA) was converted from 1 µg of RNA using the TOOLS Easy Fast RT Kit (TOOLS, TW). Levels of messenger RNA (mRNA) for *Col1a1*, *Col2a1*, *Col10a1*, *aggrecan*, *MMP13*, *ADAMT5*, *PTHrP*, *CEBPB*, *PTGS2*, *IL6*, and *TNFa* were measured by quantitative real-time PCR with the Bio-Rad iQ5 real-time PCR detection system (Bio-Rad, Hercules, CA, USA) using the TOOLS 2X SYBR qPCR Mix (TOOLS, Taiwan). Reactions took place in a 20 µL mixture containing cDNA, specific primers (Appendix A) for each gene, and the TOOLS 2X SYBR qPCR Mix. The specific PCR products were detected with the double-stranded DNA binding fluorescence dye SYBR Green [53]. Relative mRNA levels were calculated from the threshold cycle (Ct) value of each PCR product and normalized to that of GAPDH using the comparative Ct method [54]. The relative quantity of each gene’s expression in the control cells was set to 1, and all others were transformed to a ratio. A dissociation (melting) curve was generated after each PCR to check its specificity. All PCR amplifications were performed in triplicate, and the experiments were repeated at least 3 times.

### 4.8. Western Blot Analysis

HACs were lysed in radioimmunoprecipitation assay buffer (Invitrogen) supplemented with protease inhibitor cocktails (mixtures) (Sigma-Aldrich). Following lysis, proteins were separated by a NuPAGE Novex 4–12% Bis-Tris Gel (Invitrogen) and transferred to polyvinylidene difluoride nylon membranes (Sigma-Aldrich). The membranes were blocked with 5% nonfat dry milk in Tris-buffered saline and incubated overnight at 4 °C with primary antibodies against CD9 (cat. no. 602321lg, Proteintech Group, Inc., Rosemont, IL, USA; 1:1000), CD63 (256821-AP, Proteintech Group, Inc.; 1:1000), CD81 (66866-1lg, Proteintech), ALIX (cat. no. 12422-1-AP, Proteintech Group, Inc.; 1:1000), TSG101 (282831-AP, Proteintech Group, Inc.; 1:1000), and α-tubulin (cat. no. 66009-1-lg; Proteintech Group, Inc.; 1:1000). Specific bands were detected by a horseradish peroxidase–conjugated secondary antibody and were read by an enhanced chemiluminescence Western blot system from Pierce (Rockford, IL, USA).

### 4.9. ELISA

ADSC-Exos-treated HACs and the culture supernatant were collected to quantify the concentrations of extracellular matrix (type I, II, and X collagen) and SASP-related cytokines (IL6, TNFα, MMP13) by ELISA. For each assay, 100 μL of the supernatant was added to a well of the 96-well ELISA plate from the Human IL6, TNFα, MMP13, type I, II, and X collagen Express ELISA kit (cat. no. E-EL-H149, Elabscience, Inc., Houston, TX, USA; cat. no. RK00030, ABclonal, Inc. Woburn, MA, USA; cat. no. KE00078, Proteintech, Inc.; cat. no. RK01149; cat. no. RK01152, ABclonal, Inc.; cat. no. MBS722660, MyBioSource, Inc., San Diego, CA, USA) and sequentially treated with detection antibodies and chromogenic substrate according to the manufacturer’s instructions. The optical density was read at 420 nm. These protein concentrations were determined based on a standard curve that was produced using the serially diluted reference samples provided in the ELISA kit and presented as pg/mL.

### 4.10. MTS Assay after ADSC-Exos Treatment

To test the cell viability and proliferation ability after ADSC-Exos treatment, the MTS assay was used. The cells were seeded at a low cell density (1 × 10^4^/well) in a 6-well plate, treated with or without ADSC-Exos for 5 days, and the medium was changed once every two days. On the harvested day, the culture medium was removed, and 2 mL of medium containing 10 *v*/*v*% MTS reagent (CellTiter 96^®^ AQueous One Solution Cell Proliferation Assay, Promega, WI, USA) was added and allowed to react for 3.5 h. Finally, the reacted supernatant medium was collected, and 200 µL was added to a 96-well plate to measure the OD450.

### 4.11. Alcian Blue Staining

Alcian blue staining was performed using a previously described method (Luna 1960). Briefly, 1 × 10^5^ cells were seeded in a 6-well plate and cultured for 12 days. At the endpoint of culture, cells were fixed in 10% formaldehyde solution for 10 min and then washed with PBS, and 3% acetic acid solution was added for 30 min. Subsequently, cells were incubated with Alcian blue solution (pH = 2.5) (Sigma-Aldrich, Darmstadt, Germany) at RT for 24 h, and photographs were taken using a compound microscope under 100× magnification.

### 4.12. DMMB Assay to Detect Sulfated Glycosaminoglycan (sGAG) Synthesis

At every indicated time interval, HACs were collected and digested for 18 h at 60 °C using 300 μg/mL papain solution. DNA content and sulfated glycosaminoglycan (sGAG) accumulation by cells were quantified spectrofluorometrically using Hoechst 33258 dye and the Sulfate Blyscan Glycosaminoglycan Assay (Biocolor, County Antrim, UK), respectively [11,12]. The standard curve for the DMMB assay was generated using aqueous chondroitin sulfate C (Sigma-Aldrich, St. Louis, MO, USA) solution, with concentrations ranging from 0 to 25 µg/µL.

### 4.13. SA-β-Gal Staining

Senescence-associated β-galactosidase (SA-β-gal) staining was used to verify IL-1β-induced OA-like articular chondrocyte senescence. Cells were grown in six-well plates, washed, fixed, and stained with the SA-β-gal cellular senescence assay kit (Cell Biolabs) according to the manufacturer’s protocol and as previously described [55]. The sections were examined under a microscope. 

### 4.14. microRNA Extraction and Library Construction for Next-Generation Sequencing

MicroRNA was extracted from EVs enriched in a culture medium using an EV-miR Extraction Kit (Topgen Biotech., Kaohsiung City, Taiwan) following the manufacturer’s procedure and finally eluted in 20 µL of low TE buffer in each sample. The microRNA concentration was quantified by a Qubit microRNA Assay Kit on a Qubit Fluorometer (Thermo Fisher, Waltham, MA, USA). For microRNA library construction, briefly, a total of 7 µL microRNA for each sample was used to construct a microRNA library with a VAHTS Small RNA Library Prep Kit for Illumina and a VAHTS Small RNA Index Primer Kit for Illumina (Vazyme Biotech, Nanjing Shi, China) following the manufacturer’s instructions. The correct barcoding adapter-ligated library amplicon length was approximately 142 bp, and amplicons in the target region were excised from a 4% agarose gel and purified using a FastPure Gel DNA Extraction Mini Kit (Vazyme Biotech, Nanjing, China). All steps were performed according to the manufacturer’s instructions. Microfluidic electrophoresis was used to quantify the concentration and check the length distribution of the eluted barcoding adapter-ligated library (142 bp) with a DNA-2500 Kit on MultiNA MCE-202 (Shimadzu, Japan). The qualified pooled library (4 nM) was sequenced on an Illumina NextSeq 500 (single-end, 75 sequencing cycles) (Illumina, Diego, CA, USA). All steps followed the manufacturer’s user manual.

### 4.15. Bioinformatic Analysis

First, adapter sequences were removed from total reads by Clip (v1.0.3). Then, the correct insert size of approximately 15–30 bp for mature microRNAs was retained by fastp (v0.20.1) [56]. The trimmed reads were mapped with the database of human microRNA v22.1 (miRbase.org), and mapped reads of each sample were assembled and quantified using the Linux command. The target genes of highly expressed microRNAs were predicted from TargetScan v7.2 (targetscan.org), miRanda (10 March 2023, www.microrna.org), and miRDB (10 March 2023, www.mirdb.org), and the miRNA-target gene network was built by Cytoscape [57]. Finally, the union of target genes was then conducted by Gene Ontology (GO) and Kyoto Encyclopedia of Genes and Genomes (KEGG) pathway enrichment analysis by ClusterProfiler (v 3.18.1) (*p* value < 0.05, q value < 0.05) [58], and ggplot2 (v 3.3.3) was used to visualize the enrichment plots.

### 4.16. ACLT-OA Induction and ADSC/ADSC-Exo Treatment

The animal experiments were approved by the Animal Care and Use Committee of Kaohsiung Medical University (IACUC number 109242). Thirty 8-week-old male Sprague-Dawley rats were purchased from BioLASCO Taiwan and housed under standard laboratory conditions (24 °C, 12 h light-dark cycle) with food and water ad libitum. An anterior cruciate ligament transection (ACLT) model was induced as previously described [59]. Briefly, rats were randomly and evenly assigned into four groups by body weight. The experimental design has four groups, with 7–8 animals in each group: the sham operation as Sham control group (intra-articular (IA) injection PBS (40 µL) per week, n = 7), the OA+PBS group (IA injection PBS (40 µL) per week, n = 7), the OA+Exo (IA injection ADSC-Exo 7.5 × 10^9^ particles/mL in PBS (40 µL) per week, n = 8), and the OA+ADSCs (IA injection 1 × 10^6^ ADSCs in PBS (40 µL) only in the first week, n = 8 group). Each rat was anesthetized with zoletil:rompun = 4:1, and after being shaved, the right knee joint was exposed through a medial parapatellar approach, and the left knee was used as the contralateral control. The patella was dislocated laterally, and the knee was placed in full flexion, followed by ACL transection with microscissors. After surgery, the joint capsule and skin were sutured using Vicryl 4-0 (UNIK, New Taipei City, Taiwan) nonabsorbable sutures. In sham-operated animals, wounds were closed after subluxation of the patella. All groups could move freely in plastic cages until they were euthanized by CO_2_ gas overdose inhalation at 7 weeks post-surgery.

### 4.17. Weight-Bearing Distribution Test

The effect of joint damage on weight distribution through the rats’ knees was measured using a dual-channel weight average (Singa Technology, Taipei, Taiwan), which independently measured the weight-bearing ability of each hind paw. Changes in hind paw weight distribution between the right (OA) and left (contralateral control) limbs were utilized as an index of joint discomfort in the OA knee. Briefly, the weight-bearing tests were first performed before ACLT surgery and each week thereafter until the rat was euthanized. Rats were placed in an angled Plexiglas chamber positioned so that each hind paw rested on a separate force plate. The force exerted by each hind limb was averaged over a 5 s period, and each data point was the mean of three 5 s readings. The change in hind paw weight distribution was calculated by determining the percent difference in the amount of weight between the left and right limbs [59].

### 4.18. Histology and Safranin O–Fast Green Staining

Rat tibial plateaus with articular cartilage were collected and fixed with 10% neutral buffered formalin prior to histologic preparation. The decalcified samples were paraffin embedded, and 5 µm microsections of the coronary plane were prepared. GAG was stained with Safranin O–fast green (1% Safranin O counterstained with 0.75% hematoxylin and then 1% fast green; Sigma-Aldrich), and the total and red-stained areas in the articular cartilage of each proximal tibia were measured using the Image-Pro Plus software, version 5.0. The ratio of the red-stained area to the total area (red:total) in each group was calculated.

### 4.19. Immunohistochemistry

For immunohistochemistry staining, the tibial articular sections were rehydrated, and endogenous peroxidase in the tissue was blocked with 3% hydrogen peroxide. Sections were then blocked with 10% FBS for 1 h and incubated with primary antibodies against type II collagen (1:2000; mouse monoclonal antibody; Chemi-Con, Temecula, CA, USA), IL1-β (1:500; rabbit polyclonal antibody; Merck, KGaA, Darmstadt, Germany), ADAMTS5 (1:1000; rabbit polyclonal antibody; Novus Biologicals, LLC, Centennial, CO, USA), and MMP13 (1:200; rabbit polyclonal antibody; Bioss, MA, USA) at 4 °C overnight. Then, the samples were incubated for 30 min with biotin-labeled goat anti-mouse/rabbit immunoglobulin and horseradish peroxidase-conjugated streptavidin (BioCare Medical, Pacheco, CA, USA) secondary antibodies. Staining with a 3,3′-diaminobenzidine solution (Dako) containing 0.01% hydrogen peroxide resulted in a brown color. Finally, sections were counterstained with hematoxylin and observed under a microscope. The relative density of immunostaining (density/area; mean ± SEM area 25.44 ± 2.77 mm^2^) was measured using the Image-Pro Plus software, version 5.0.

### 4.20. Statistical Analysis

Data are presented as the mean and SEM of results from the in vitro study and the in vivo study. All experiments were repeated at least 3 times. Statistical significance was evaluated by a one-way analysis of variance, and multiple comparisons were performed using Scheffe’s test. *p* values less than 0.05 were considered significant.

## Figures and Tables

**Figure 1 ijms-24-13414-f001:**
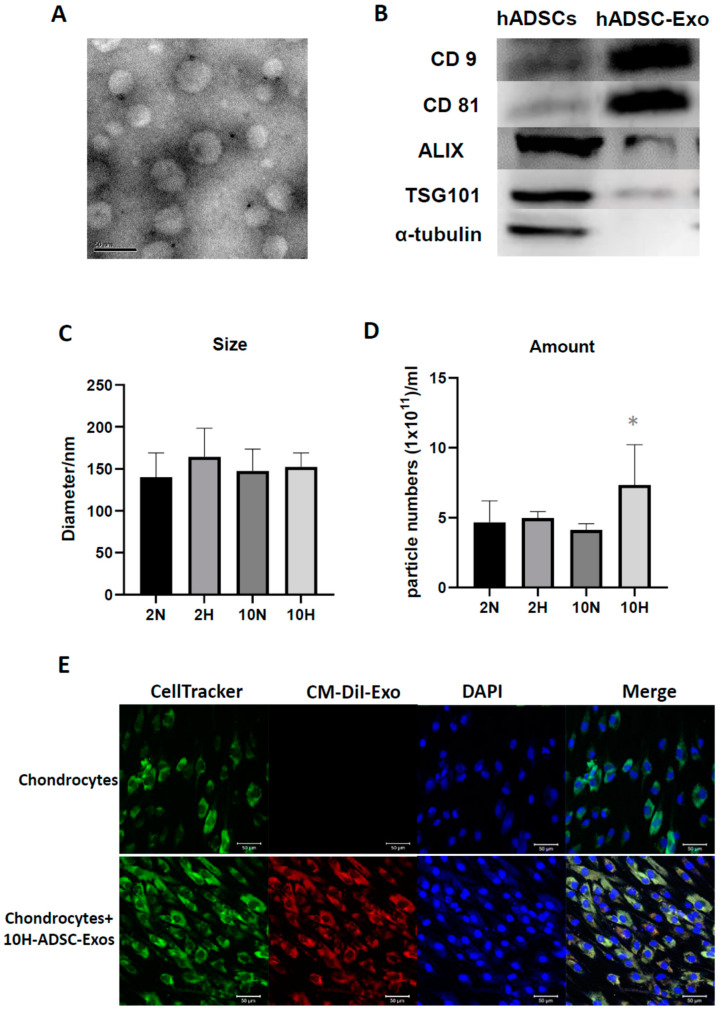
Exosomes induced by normoxia/hypoxia and normal/low serum culture conditions in cultured ADSCs. (**A**) Transmission electron microscopy (TEM) image of ADSC-Exos (scale bar: 50 nm). (**B**) Western blot analysis of the protein levels of ADSC-Exo markers in ADSCs and ADSC-Exos was performed, including CD9, CD63, CD81, ALIX, TSG101, and α-tubulin levels, which were used as negative controls. (**C**) Nanoparticle tracking analysis (NTA) evaluating the ADSC-Exo size and (**D**) concentration after four conditions induced ADSCs, including 20% O_2_ (normaxia) + 2% de-Exo-FBS, (2 N), 1% O_2_ (hypoxia) + 2% de-Exo-FBS (2H), 20% O_2_ (normaxia) + 10% de-Exo-FBS (10 N), and 1% O_2_ (hypoxia) + 10% de-Exo-FBS (10H) for 2 days. Data represent the mean exosome size, and each column represents the mean ± SEM of six replicate cultures. *, *p* < 0.05 compared to 2 N induction. (**E**) 10H-ADSC-Exo uptake by HACs. The CM-DiI-labeled exosomes (red fluorescence-stained) were taken up by cell tracker-labeled chondrocytes (green fluorescence-stained; DAPI-stained nucleus: blue fluorescence) on Day 12.

**Figure 2 ijms-24-13414-f002:**
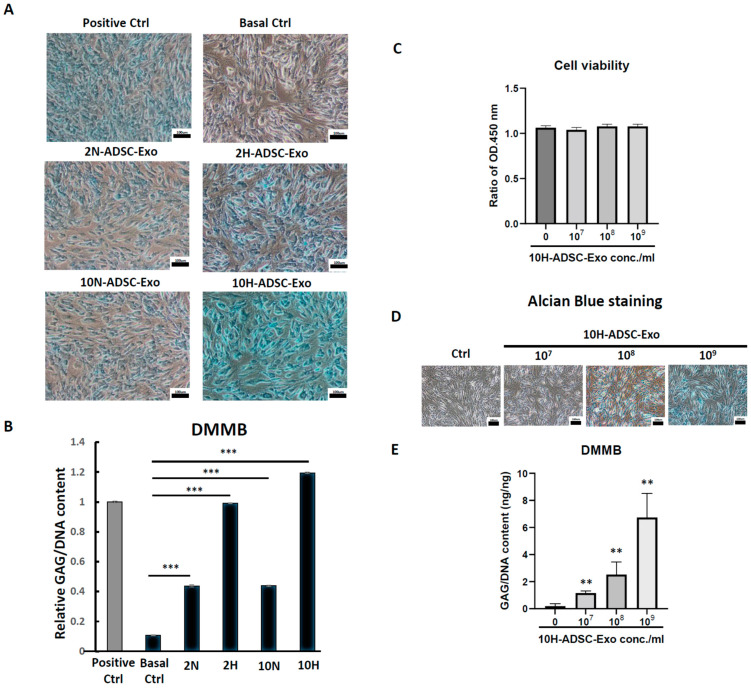
We compared different culture conditions and found the optimal function of exosomes derived from ADSCs on chondrocytes. (**A**) Alcian blue staining and (**B**) DMMB assay analysis of GAG synthesis in HACs treated with normal culture medium (as a positive control), basal medium (only DMEM), and 4-condition-induced ADSC-Exos (2 N, 2H, 10 N, 10H) supplemented in basal medium. Data represent the relative GAG/DNA content, and each column represents the mean ± SEM of six replicate cultures. ***, *p* < 0.001 compared to the basal control culture. (**C**) MTS assay evaluating chondrocyte viability after treatment with 10H-ADSC-Exos (10^7^~10^9^ particles/mL) for 5 days. Data represent the relative cell viability, and each column represents the mean ± SEM of six replicate cultures. There was no difference compared with the basal control culture (Ctrl). (**D**) GAG synthesis of HACs treated with different doses of 10H-ADSC-Exos (10^7^~10^9^ particles/mL) by Alcian blue staining and (**E**) DMMB assay. Data represent the relative GAG/DNA content, and each column represents the mean ± SEM of six replicate cultures. **, *p* < 0.01, compared to the basal control culture.

**Figure 3 ijms-24-13414-f003:**
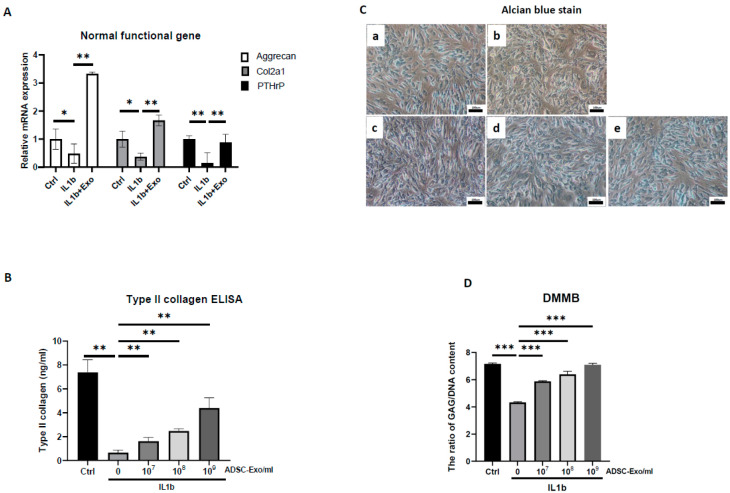
Hypoxia-induced ADSC-Exos enhanced cartilaginous matrix synthesis in IL-1β-induced OA-like inflammaging articular chondrocytes. (**A**) The mRNA expression of the normal functional genes aggrecan, col2a1, and PTHrP in vitro in HACs treated with IL1-β with or without ADSC-Exos (10^9^ particles/mL) for 12 days. The mRNA expression of GAPDH in HACs was used as the internal control. (**B**) Type II collagen ELISA was performed to quantify the protein level when HACs were treated with IL1-β and with or without ADSC-Exos (10^7^~10^9^ particles/mL) for 12 days. (**C**) Alcian blue staining ((**a**) Ctrl, (**b**) IL-1β, (**c**) IL-1β+ADSC-Exo 10^7^ particles/mL, (**d**) IL-1β+ADSC-Exo 10^8^ particles/mL, (**e**) IL-1β+ADSC-Exo (10^9^ particles/mL) and (**D**) DMMB assay analysis of GAG synthesis in HACs treated with IL1-β and with or without ADSC-Exo (10^7^~10^9^ particles/mL) for 12 days. Each column represents the mean ± SEM of six replicate cultures. Data from each group were compared with those of the IL-1β only group, *, *p* < 0.05, **, *p* < 0.01, ***, *p* < 0.001.

**Figure 4 ijms-24-13414-f004:**
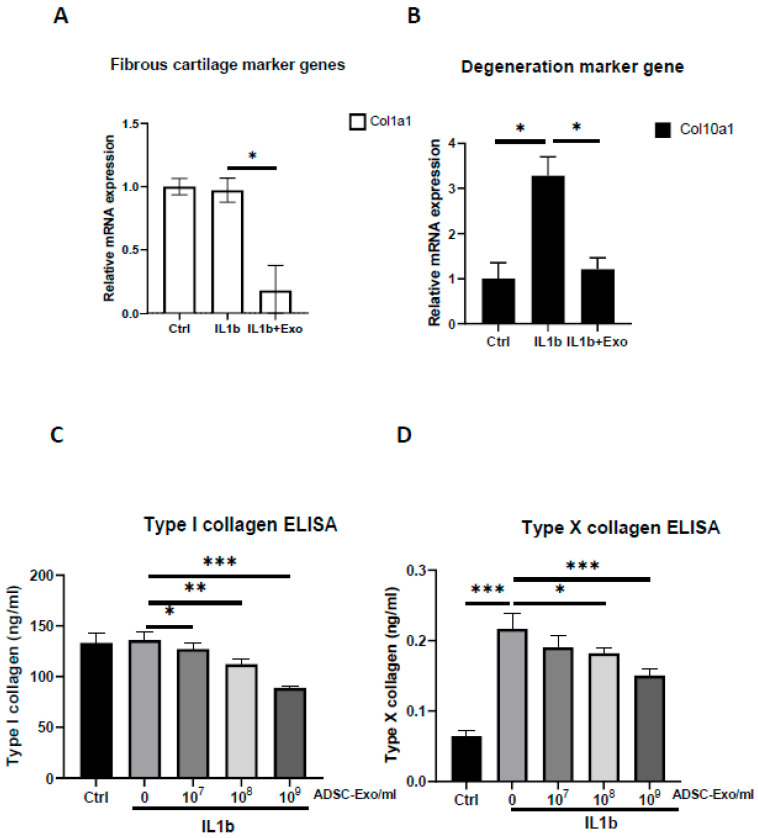
Hypoxia-induced ADSC-Exos inhibited fibrous and degenerated cartilage in IL-1β-induced OA-like articular chondrocytes. (**A**) Fibrous cartilage genes, col1a1, and (**B**) degeneration marker gene, col10a1, mRNA expression in vitro in HACs treated with IL1-β and with or without ADSC-Exos (10^9^ particles/mL) for 12 days. GAPDH mRNA was used as the internal control. (**C**) Type I and (**D**) type X collagen ELISAs were performed to quantify the protein levels when HACs were treated with IL1-β and with or without ADSC-Exos (10^7^~10^9^ particles/mL) for 12 days. Each column represents the mean ± SEM of six replicate cultures. Data from each group were compared with those of the IL-1β only group, *, *p* < 0.05; **, *p* < 0.01; ***, *p* < 0.001.

**Figure 5 ijms-24-13414-f005:**
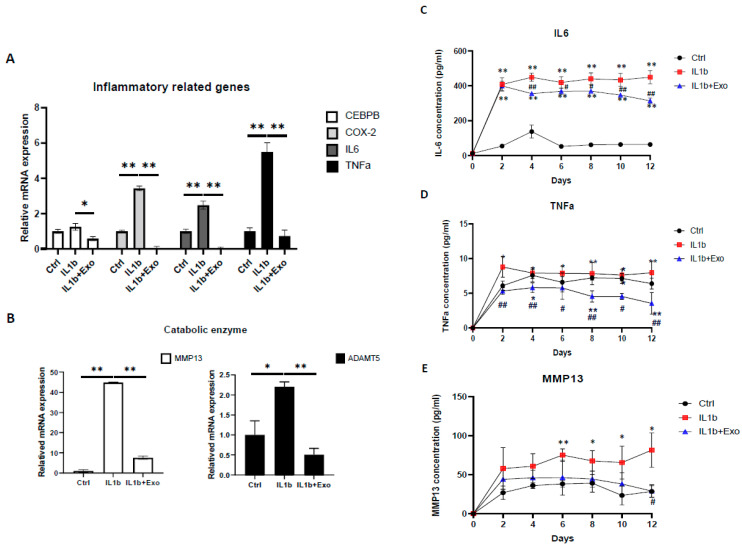
Hypoxia-induced ADSC-Exos suppressed inflammatory cytokines and degradation enzymes in inflammaging articular chondrocytes. (**A**). Inflammatory-related gene (CEBPβ, COX-2, IL-6, and TNF-α) and (**B**) catabolic gene (MMP13 and ADAMT5) mRNA expression in HACs treated with IL1-β with or without ADSC-Exos (10^9^ particles/mL) for 12 days in vitro. GAPDH mRNA expression was used as the internal control. Data from each group were compared with those of the IL-1β only group, *, *p* < 0.05; **, *p* < 0.01. (**C**) IL-6, (**D**), TNF-α, (**E**), and MMP13 ELISAs were performed to quantify the protein levels when HACs were treated with IL1-β with or without ADSC-Exos (10^9^ particles/mL) for 12 days. Each column or scatter plot represents the mean ± SEM of six replicate cultures. Data from each group were compared with those of the control group, *, *p* < 0.05; **, *p* < 0.01 and were compared with those of the IL-1β group, #, *p* < 0.05; ##, *p* < 0.01.

**Figure 6 ijms-24-13414-f006:**
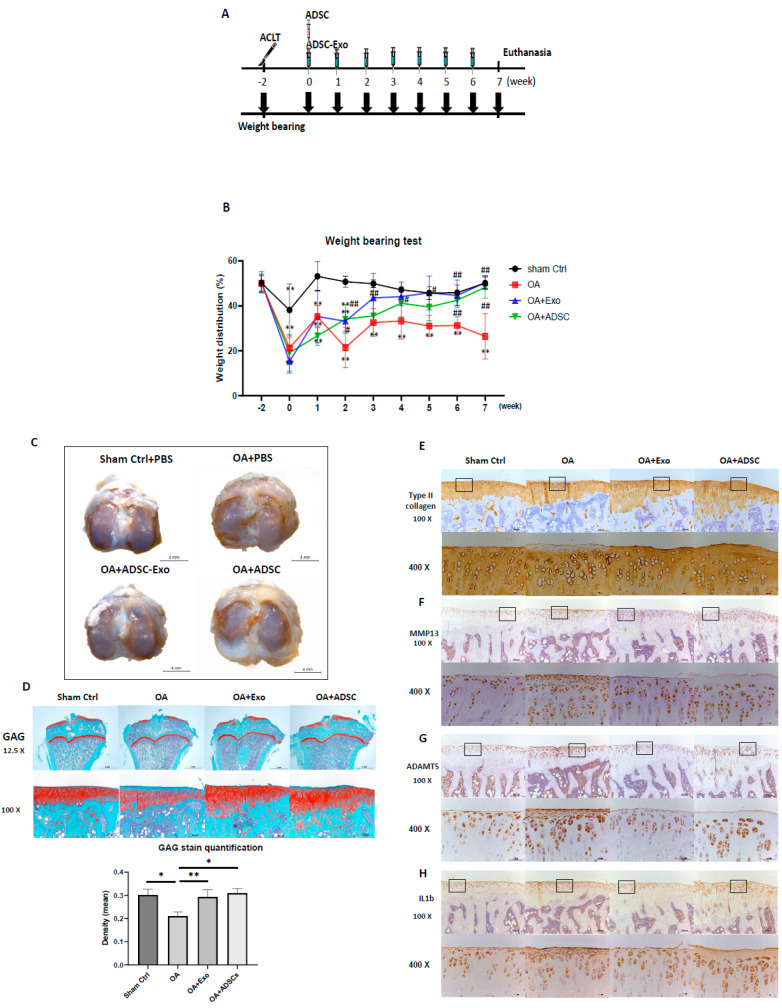
Hypoxia−induced ADSC−Exos and ADSC treatment ameliorate ACLT−OA in rats. (**A**) Schematic of the experimental protocol. The rats were followed for 8 weeks after ACLT or sham surgery, and static weight-bearing was assessed before ACLT and every week post-surgery. (**B**) Comparison of the weight distribution analysis of ADSCs or ADSC-Exo treatment in ACLT-OA rats using the weight-bearing test. Sham control and OA control group rats received PBS in the right knee. Each scatter plot presents the means ± SEM of 7 to 8 replicate cultures. Data from each group are compared with those of the sham control group (** *p* < 0.01) or with those of the OA group (# *p* < 0.05, ## *p* < 0.01). (**C**) Gross evaluation of tibia in knee joints. The photograph represents the surfaces of tibial plateaus (scale bar: 4 mm). (**D**) Safranin O staining analysis of GAG content. The upper panel shows the Safranin O staining of tibia sections (scale bars: 2 mm), and the second panel shows the higher magnification of the boxed area (scale bar: 200 µm). Comparison of the density of the Safranin O stained area (red color) in articular cartilage of tibia sections in all groups after ADSC and ADSC-Exo treatment. Each column shows the mean and SEM of 7–8 samples. *, *p* < 0.05, **, *p* < 0.01 compared to the OA group. Immunohistochemical analysis of type II collagen (**E**), MMP13 (**F**), ADAMT5 (**G**), and IL1-β (**H**) in articular cartilage of tibia sections in the Sham Ctrl, OA, OA+ADSC-Exo and OA+ADSCs groups. The upper panel shows the IHC staining of ADAMT5, type II collagen, MMP13, and IL1-β (scale bar: 200 µm), and higher magnification of the box area (Scale bar: 50 µm) is shown in the bottom in all groups.

**Figure 7 ijms-24-13414-f007:**
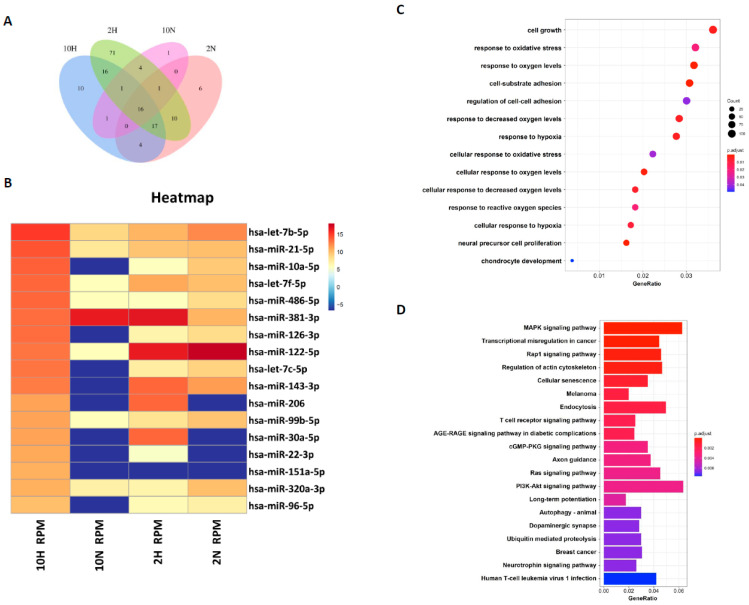
Next generation sequencing (NGS) and bioinformatic analysis of miRNA profiles in 4 culture conditions induced by ADSC-Exos. (**A**) Comparison and intersection of miRNAs from 4 culture conditions induced ADSC−Exos (2H, 2N, 10H, 10N group). (**B**) Heatmap of the differentially expressed miRNAs from 4 groups. Hierarchically represented the miRNAs from the 10N group and compared them with the other groups (*p* value < 0.05 and fold change > 2). (**C**) GO and (**D**) KEGG pathway analyses. There were 3358 target genes of 7 selected miRNAs in ADSC-Exos.

## Data Availability

MDPI Research Data Policies.

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
