# Peer review of "Exosomes Derived from Hypoxia-Cultured Human Adipose Stem Cells Alleviate Articular Chondrocyte Inflammaging and Post-Traumatic Osteoarthritis Progression"

_ijms, 2023, doi:10.3390/ijms241713414_

Round 1

Reviewer 1 Report

Comments:

This study examines the effect of exosomes derived from hypoxia-cultured human adipose stem cells on articular chondrocyte inflammation, aging and osteoarthritis progression in vitro and in vivo in a rat model of ACLT. The authors have represented their data in a detailed manner. This study, although trying to answer an interesting question, has some issues that need to be addressed. Some of the minor concerns currently with the manuscript are listed below.

The experimental design appears robust and well-structured. The use of in vitro models with IL1-β-induced human articular chondrocytes (HACs) and an in vivo model of ACLT-induced OA rats allows for a comprehensive assessment of the effects of hypoxia-ADSC-Exos on various facets of OA progression. This combination of in vitro and in vivo approaches strengthens the validity of the findings. The study provides valuable mechanistic insights into the effects of hypoxia-ADSC-Exos on OA-related processes. The observed effects on cartilage markers (GAG, type II collagen), matrix degradation enzymes (MMP13, ADAMT5), and inflammatory cytokines (TNFα, IL-6) in OA-like HACs suggest that hypoxia-ADSC-Exos may influence key pathways contributing to OA pathology.

The utilization of Next-Generation Sequencing (NGS) and bioinformatic analysis to identify potential miRNAs adds depth to the study. However, the functional validation of these miRNAs and their specific targets is necessary to establish a causal link between miRNA regulation and observed effects.

The optimal dosage and treatment duration of hypoxia-ADSC-Exos in the ACLT rat model needs further investigation. Additionally, the study does not provide information on the long-term effects of repeated intra-articular administrations, which is important for clinical relevance. Justifications from literature is required.

A detailed description of sample sizes and statistical power calculations for both in vitro and in vivo experiments would enhance the rigor of the study and strengthen the interpretation of the results.

The study primarily focuses on hypoxia-ADSC-Exos and its effects. A more comprehensive analysis that compares the effects of hypoxia-ADSC-Exos with exosomes from other sources or treatment approaches would provide a more complete picture of their therapeutic potential.

While the study shows promise for future OA therapy, it is important to note that additional preclinical studies and, ultimately, well-controlled clinical trials are required to determine the safety and efficacy of hypoxia-ADSC-Exo treatment in humans.

Line 484 What is deExo-FBS? If Exos contain 2% or 10% FBS, then how do we evaluate their effects if FBS already has commercialized exosomes or we use FBS-deprived exosomes?

Line 511 ADSC-Exo solution was diluted with basal medium for in vitro treatment; what is the ratio of Exo solution and basal medium? What part of Exo solution can you dissolve in basal medium?

Line 514 Why did you select only a 12-day time point for ADSC-Exo treatment after inducing HACs?

Nil 

Author Response

Reviewer #1:

Comments:

This study examines the effect of exosomes derived from hypoxia-cultured human adipose stem cells on articular chondrocyte inflammation, aging and osteoarthritis progression in vitro and in vivo in a rat model of ACLT. The authors have represented their data in a detailed manner. This study, although trying to answer an interesting question, has some issues that need to be addressed. Some of the minor concerns currently with the manuscript are listed below.

The experimental design appears robust and well-structured. The use of in vitro models with IL1-β-induced human articular chondrocytes (HACs) and an in vivo model of ACLT-induced OA rats allows for a comprehensive assessment of the effects of hypoxia-ADSC-Exos on various facets of OA progression. This combination of in vitro and in vivo approaches strengthens the validity of the findings. The study provides valuable mechanistic insights into the effects of hypoxia-ADSC-Exos on OA-related processes. The observed effects on cartilage markers (GAG, type II collagen), matrix degradation enzymes (MMP13, ADAMT5), and inflammatory cytokines (TNFα, IL-6) in OA-like HACs suggest that hypoxia-ADSC-Exos may influence key pathways contributing to OA pathology.

The utilization of Next-Generation Sequencing (NGS) and bioinformatic analysis to identify potential miRNAs adds depth to the study. However, the functional validation of these miRNAs and their specific targets is necessary to establish a causal link between miRNA regulation and observed effects.

The optimal dosage and treatment duration of hypoxia-ADSC-Exos in the ACLT rat model needs further investigation. Additionally, the study does not provide information on the long-term effects of repeated intra-articular administrations, which is important for clinical relevance. Justifications from literature is required.

A detailed description of sample sizes and statistical power calculations for both in vitro and in vivo experiments would enhance the rigor of the study and strengthen the interpretation of the results.

The study primarily focuses on hypoxia-ADSC-Exos and its effects. A more comprehensive analysis that compares the effects of hypoxia-ADSC-Exos with exosomes from other sources or treatment approaches would provide a more complete picture of their therapeutic potential.

While the study shows promise for future OA therapy, it is important to note that additional preclinical studies and, ultimately, well-controlled clinical trials are required to determine the safety and efficacy of hypoxia-ADSC-Exo treatment in humans.

Response: Thanks for the valuable comments! Further studies of comparisons with different sources of Exosomes and pre-clinical studies for the optimal doses and long-term treatment effect for OA are definitely important to be investigated.

Line 484 What is deExo-FBS? If Exos contain 2% or 10% FBS, then how do we evaluate their effects if FBS already has commercialized exosomes or we use FBS-deprived exosomes?

Response: Thanks for your questions. The deExo-FBS (Thermo Fisher Gibco™, USA) is an exosome-depleted fetal bovine serum (FBS) that has been specifically designed for researchers interested in isolating exosomes from cultured cells without the complications introduced by bovine exosomes present in conventional FBS. Therefore,

in our experiments for testing the effect of “Exos contain 2% or 10% FBS”, we used deExo-FBS.

Line 511 ADSC-Exo solution was diluted with basal medium for in vitro treatment; what is the ratio of Exo solution and basal medium? What part of Exo solution can you dissolve in basal medium?

Response: ADSC-Exos were isolated from the CM by ultracentrifugation and resuspended in 1 mL of sterilized PBS, and the final concentration of ADSC-Exos is around 5x 1011/ml. ADSC-Exo solution used in the experiments was between 107 and 109 particles/mL; therefore, ADSC-Exos diluted in the basal medium over 100 times.

Line 514 Why did you select only a 12-day time point for ADSC-Exo treatment after inducing HACs?

Response: At the beginning of this study, we had some preliminary results that showed gradual improvement after ADSC-Exo treatment, and the best effect on cartilage matrix formation was at 12 days. Therefore, we chose the 12-day time point to show our results.

Reviewer 2 Report

This is a nice study about exosomes derived from hypoxia-cultured human adipose stem cells alleviate articular chondrocyte inflammaging and osteoarthritis progression. I suggest it for publication after the following points are addressed.

1. Line 52-55, one recent review (Advanced Materials 33 (18), 2005513) should be included to support such a claim.

2. Figure 1c, Y axis: 'Diameter mode' to 'Diameter'.

3. The resolution of figure 1 should be improved.

4. The scale bar in figure 1E is not clear.

5. Line 533-539, cryo-TEM is more suitable to characterize EXOs.

6. The authors should add OARSI scores before and after the treatments to show the condition of cartilage injury that happened in joints.

Minor editing of English language required

Author Response

Thanks for the reviewer’s valuable comments! The point-to-point responses are as following:

Reviewer #2:

Comments and Suggestions for Authors:

This is a nice study about exosomes derived from hypoxia-cultured human adipose stem cells alleviate articular chondrocyte inflammaging and osteoarthritis progression. I suggest it for publication after the following points are addressed.

  1. Line 52-55, one recent review (Advanced Materials 33 (18), 2005513) should be included to support such a claim.

Response: We have added the review article (Advanced Materials 33 (18), 2005513) in the Introduction (Line 56).

  1. Figure 1c, Y axis: 'Diameter mode' to 'Diameter'.

Response: We have changed the 'Diameter mode' to 'Diameter' in Y axis of Figure 1c, as reviewer’s comment.

  1. The resolution of figure 1 should be improved.

Response: We have improved the resolution of figure 1, as reviewer’s comment.

  1. The scale bar in figure 1E is not clear.

Response: We have bolded the scale bar and made it clearer in figure 1E, as reviewer’s comment.

  1. Line 533-539, cryo-TEM is more suitable to characterize EXOs.

Response: There are two types of electron microscopy, Cryo-transmission electron microscopy (cryo-TEM) and transmission electron microscopy (TEM), are widely used to characterize and visualize exosomes. [Szatanek R, Baj-Krzyworzeka M, Zimoch J, Lekka M, Siedlar M, Baran J. (2017) The Methods of Choice for Extracellular Vesicles (EVs) Characterization. Int J Mol Sci 18(6).] We chose the TEM that we are able to use in our facility.

  1. The authors should add OARSI scores before and after the treatments to show the condition of cartilage injury that happened in joints.

Response:  For OARSI of rat OA, there is guidance from “Title: The OARSI histopathology initiative – recommendations for histological assessments of osteoarthritis in the rat. Osteoarthritis and Cartilage Volume 18, Supplement 3, October 2010, Pages S24-S34”. According to this paper, the semiquantitative OARSI scoring includes two parts, macroscopic scoring of gross tibial plateaus and microscopic scoring of the cartilage matrix loss by histology staining section (e.g., GAG staining). In this study, our ACLT model developed a mild OA, so the macroscopic scoring of this study showed a non-significant difference. Therefore, we used the GAG quantification of cartilage to present the microscopic change instead of using a semi-quantification of scoring. The results are shown in Fig 6D.

Reviewer 3 Report

The authors have produced an interesting and useful study that provides an update on the role of hypoxia-cultured ADSC-secreted exosomes (hypoxia-ADSC-Exos) as generating a beneficial effect on the normal functions of human articular chondrocytes (HACs), and which could attenuate the senescence-associated secretory phenotype (SASP) of OA-like HACs in vitro and further suppress the progression of osteoarthritis in rats.

However, I would like to make some observations before recommending your work for publication.

1. Could the authors add the type of study in the title?

2. Could you please detail the approval of the Ethics Committee in the corresponding section?

3. There is an interesting paper developing to identify differences in blood investigations between total hip replacement and total knee replacement with respect to inflammatory mediators. I recommend the authors to discuss it: 10.1097/TGR.00000000000000000337

4. Could the authors provide a graphical abstract of the study?

5. In the introduction /Discussion, I recommend that the authors to comment on the conservative treatment that can be very useful in mild to moderate OA. 

6. Could you add a section on "Clinical Implications"?

Author Response

Reviewer #3:

Comments and Suggestions for Authors:

The authors have produced an interesting and useful study that provides an update on the role of hypoxia-cultured ADSC-secreted exosomes (hypoxia-ADSC-Exos) as generating a beneficial effect on the normal functions of human articular chondrocytes (HACs), and which could attenuate the senescence-associated secretory phenotype (SASP) of OA-like HACs in vitro and further suppress the progression of osteoarthritis in rats.

However, I would like to make some observations before recommending your work for publication.

Thanks for the reviewer’s valuable comments! The point-to-point responses are as following:

  1. Could the authors add the type of study in the title?

Response: We have changed the title to ‘’Exosomes Derived from Hypoxia-Cultured Human Adipose Stem Cells Alleviate Articular Chondrocyte Inflammaging and Post-traumatic Osteoarthritis Progression‘’. Does that meet the reviewer’s comment of “the type of study”?

  1. Could you please detail the approval of the Ethics Committee in the corresponding section?

Response: We have added the statement and provided the certificate ID of the approval of IACUC in KMU (Page, Line).

  1. There is an interesting paper developing to identify differences in blood investigations between total hip replacement and total knee replacement with respect to inflammatory mediators. I recommend the authors to discuss it: 10.1097/TGR.00000000000000000337

Response: ‘Topics in Geriatric Rehabilitation Volume 37, Number 4, 227-229 : 10.1097/TGR.0000000000000337 Title: Impact on Blood Tests of Lower Limb Joint

Replacement for the Treatment of Osteoarthritis.’ In this article, the authors did not find any statistically significant difference between pre- and postoperative hematologic variables between hip or knee replacement groups. They recommend in the future the clinical implications should be performed with particular attention to inflammatory indices in routine blood investigation. Generally, total hip and total knee replacement surgery are for patients with severity of joint problems and advanced symptomatic arthritis. In this study, our ACLT model developed a mild OA in rats, and the inflammatory response was mild and mainly limited in OA cartilage. Therefore, discussing inflammatory indices in routine blood investigations is not the focus of this paper.

  1. Could the authors provide a graphical abstract of the study?

Response: We have provided a graphical abstract of this study, as the reviewer’s comment.

  1. In the introduction /Discussion, I recommend that the authors to comment on the conservative treatment that can be very useful in mild to moderate OA.

Response: We have added a paragraph in Discussion (Page15. Line458-468.) as following:

‘‘To look back the traditional treatments for OA include management of pain and inflammation by anti-inflammatory drugs or even corticosteroid injection in some several cases. At the end-stage of OA needs joint replacement. At the end-stage of OA needs joint replacement. In current OA treatment, despite the administration of NSAIDs, hyaluronan intra-articular injection has also been used as a viscosupplementation, but not effective for those with damaged articular cartilage [1]. For mild to moderate OA, several treatment strategies of regenerative medicine are exploring, such as using platelet-enriched plasma (PRP), cell-based therapy or acellular MSC exosomes. In the present study, we demonstrated that using hypoxia-ADSC-Exos can alleviate the degeneration of articular chondrocytes and eventually OA progression, which may be potentially a future treatment method for OA.’’

  1. Could you add a section on "Clinical Implications"?

Response: We have added a paragraph in Discussion (Page15. Line458-468.), as reviewer’s comment.

  1. Migliore, A.; Granata, M., Intra-articular use of hyaluronic acid in the treatment of osteoarthritis. Clin Interv Aging 2008, 3, (2), 365-9.

Round 2

Reviewer 1 Report

The authors addressed the review point. The manuscript is accepted

Reviewer 3 Report

The authors improved the previous version of their manuscript, so I recommend it for publication.

Congratulations

No comments